# Publication games: In the web of reciprocity

**Zoltán Barta** *

ELKH-DE Behavioural Ecology Research Group, Department of Evolutionary Zoology, University of Debrecen, Debrecen, Hungary

* barta.zoltan@science.unideb.hu

## Abstract

The present processes of research assessment, i.e. focusing on one or a few, related, scientometrics, foster questionable authorship practices, like gifting authorship to non-contributing people. An especially harmful one of these unethical practices is the formation of publication cartels, where authors offer gift authorship to each other reciprocally. Here, by developing a simple model and a simulation of the publication process I investigate how beneficial cartels can be and what measure can be used to restrict them. My results indicate that publication cartels can significantly boost members' productivity even if paper counts are weighted by the inverse of author number (the $1/n$ rule). Nevertheless, applying the $1/n$ rule generates conflicts of interest both among cartel members themselves and between cartel members and non-members which might lead to the self-purification of the academic publishing industry.

**Data Availability Statement:** Computer code needed to reproduce this study is available from GitHub (https://github.com/zbartab/PaperPump).

**Funding:** ZB was supported by the Thematic Excellence Programme (TKP2020-IKA-04) of the Innovációs és Technológiai Minisztérium, Hungary and the Thematic Excellence Programme

## Introduction

Research integrity (ethical behaviour, sound methodology and rigorous peer review, [1]) provides assurance that scientific activities lead to trustable and replicable results. Research integrity is, however, under threat as a result of how science currently operates. The recent, unprecedented expansion of science, exemplified, for instance, by the exponentially growing number of scientific articles [2], gives way to the wide-spread use of scientometry for assessing the productivity and impact of researchers [3]. As science is usually funded by public resources, the desire to measure the performance of its actors is well justified. Introducing the assessment of scientists by one or a few metrics, like number of publications or citations, together with the hyper-competitiveness of science had, however, somehow unexpected consequences [4] (not only among scholars but even among scholarly institutes [5]).

As, among others, Charles Goodhart observed, if a metric is used as a target then it becomes a bad metric [2, 6]. This happens because people, in response to introduction of a target, alter their own behaviour to affect the metric directly instead to modify the activity the change of which was intended by introducing the metric [7]. In the recent process of corporatisation of science two such metrics became relevant: the numbers of papers and citations [8].

Goodhart's law is well illustrated by the introduction of the number of papers as a measure of productivity in science. Using this measure is based on the assumption that characteristics of scientific papers (like length or number of coauthors) are fixed and hence targeting more

(TKP2021-NKTA-32) of the Nemzeti Kutatási, Fejlesztési és Innovációs Alap, Hungary. The funders had no role in study design, data collection and analysis, decision to publish, or preparation of the manuscript.

**Competing interests:** The authors have declared that no competing interests exist.

papers automatically leads to the generation of more new knowledge. Unfortunately, this was not what had happened, scientists responded in some unexpected, nevertheless clearly rational but sometimes unethical, ways [9, 10]. For instance, they reduced the length of papers [2], i.e. they are publishing the same amount of knowledge in more papers (salami articles). Furthermore, mangling with authorship appeared where offering authorship to those who did not contributed to the given paper considerably (honorary authorship) can quickly increase their number of publications, again without any increase in knowledge produced [3, 4, 9, 10]. A possible sign of this questionable authorship practice can be the recent raise of number of authors per paper [2]. One may argue that more authors per paper is the sign of science becoming more interdisciplinary. A recent analysis is, however, unlikely to support this conclusion; the number of coauthors increases with time even after controlling for attributes related to complexity of science [11]. Another reason for the increased number of coauthors might be the increased efficiency that can follow from the increased possibility for division of labour facilitated by more authors [12]. In this case, however, it is expected that the number of papers per author also increases, which seems not to be the case [2].

Questionable authorship practice, on the other hand, appears to be common. Recent surveys suggest that about 30% of authors were involved in these unethical practices [4, 8–10, 13, 14]. One of these practices is ghost authorship when someone who has significantly contributed to the article is excluded from the author bylist [15]. In other forms (honorary authorship) just the opposite happens; those are offered authorship who have not (considerably) contributed to the work published [9, 10]. Several reasons can be behind gifting authorship to someone. Junior authors might include more senior ones because of respect or they are forced to do so [16]. Senior authors may gift authorship to juniors in order to help them obtain postdoctoral scholarships or tenure [17].

A very efficient way to increase the number of publications may be to practice honorary authorship reciprocally. The most organised form of this behaviour is founding publication cartels. The cartel is formed by a group of people who agree to mutually invite each others to their own publications as guest authors without any contribution. As in recent assessment practice coauthored papers count as a whole publication to every coauthor on the bylist, publication cartels can significantly boost the productivity of cartel members. This is the phenomenon which is called as 'publication club' by [12]. As the noun of 'club' involves a positive connotation I prefer to use 'cartel' for this under studied but highly unethical behaviour. Simple argument suggests that sharing the credit of a publication among the coauthors can decrease the incentive of forming cartels [12]. The simplest scenario for sharing is the $1/n$ rule under which only $1/n$ part of a publication is attributed to each of the $n$ coauthors of the given paper [12].

In this paper I develop a simple model of publication cartels to understand how effective they are to increase members' productivity and whether it is possible to eliminate them by applying different measures, like the $1/n$ rule. I then extend my study to situations resembling more to real world conditions by developing a computer simulation of cartels. I use this simulation to investigate how using different metrics of productivity affect authors outside of cartels.

## The model

We compare the publication performance of two authors, author $A_1$ and author $B_1$. Authors work in separate groups (group $A$ and group $B$ respectively) each of which contains $G_i$ ($i = A$, $B$) people (including the focal author). Each author in group $A$ produces $p_A$ papers in a year by collaborating with $c_A$ authors from outside of the group, i.e. their primary production is $p_A$.

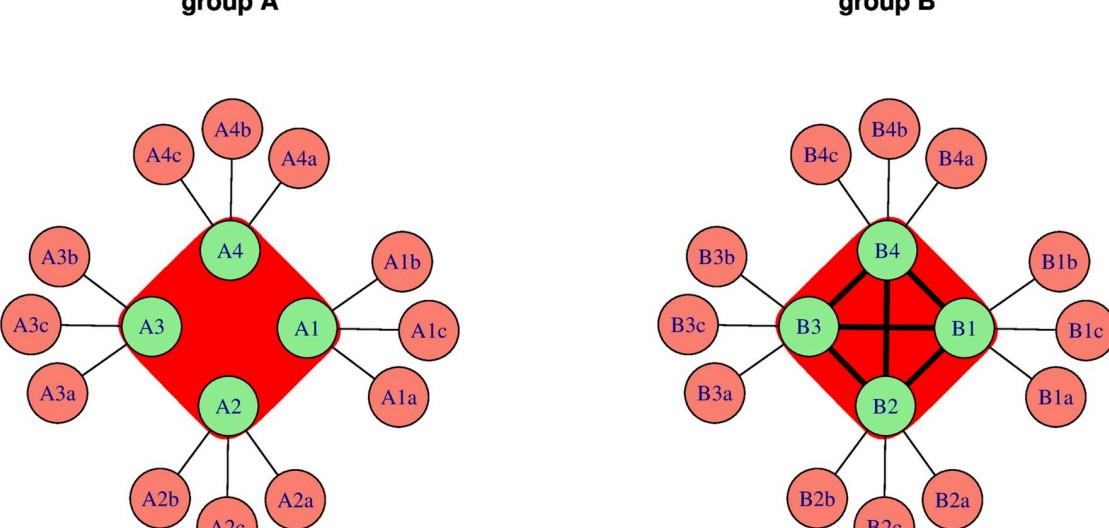

**Fig 1. The publication relationships in groups *A* and *B* of the model.** Nodes are authors, while edges symbolise shared publications. Groups of four authors are marked by the underlying shapes. In group *A* authors work with several coauthors from outside of the group but they do not invite group mates to be coauthors on their own papers. Contrarary, authors in group *B* form a publication cartel i.e. each author invites all other authors in the group to be a coauthor (note the connections between group members).

Similarly, each author in group *B* primarily produces $p_B$ papers by collaborating with $c_B$ people outside of the group. The difference between authors $A_1$ and $B_1$ is that authors in group *A* work independently of each other, while authors in group *B* invite all other group members to be a coauthor on their papers independently of their contribution to that paper (Fig 1). In other words, authors in group *B* form a publication cartel.

For simplicity, we assume that $G_A = G_B = G$, $(G > 1)$, $p_A = p_B = p$ and $c_A = c_B = c$, i.e. author groups are of the same size, authors produce the same number of primary papers and they have the same number of coauthors from outside of the group. In this case the total numbers of papers produced by the groups, the group productivity, are equal ($Gp = G_A p_A$ and $Gp = G_B p_B$, respectively). The total numbers of papers (co)authored by authors $A_1$ and $B_1$ are, however, different. Author $A_1$ writes $n_A = p_A = p$ papers. On the other hand, author $B_1$ (co)authors $n_B = p_B + (G_B - 1)p_B = G_B p_B = Gp$ papers. In the case of author $B_1$ the term $(G_B - 1)p_B$ represents the papers on which author $B_1$ is invited as honorary author. It is easy to see that as far as $G > 1$ author $B_1$ will have many more paper than author $A_1$, i.e. $n_B > n_A$.

A natural way to correct for this bias is to taking into account the number of authors each paper has and instead of counting the papers themselves as a measure of productivity one sums the inverse of the number of authors (weighted number of papers or the $1/n$ rule, [12, 18]):

$$w = \sum_{i=1}^{n} \frac{1}{1 + C}.$$

Here, number 1 in the denominator symbolises the focal author, while $C$ is the number of coauthors. For author $A_1$, $C = c_A = c$. On the other hand, for author $B_1$, $C = (G_B - 1) + c_B = (G - 1) + c$. If $c = 0$, then the division by the number of coauthors works, we regain the number of papers the authors produced without inviting their group members.

For author $A_1$:

$$w_A = \sum_{i=1}^{n_A} \frac{1}{1} = \sum_{i=1}^{n_A} 1 = n_A = p.$$

For author $B_1$:

$$w_B = \sum_{i=1}^{n_B} \frac{1}{1 + G - 1} = \sum_{i=1}^{Gp} \frac{1}{G} = \frac{Gp}{G} = p.$$

On the other hand, if the focal authors collaborate with others outside of their groups, as Fig 1 illustrates, the situation changes (Fig 2):

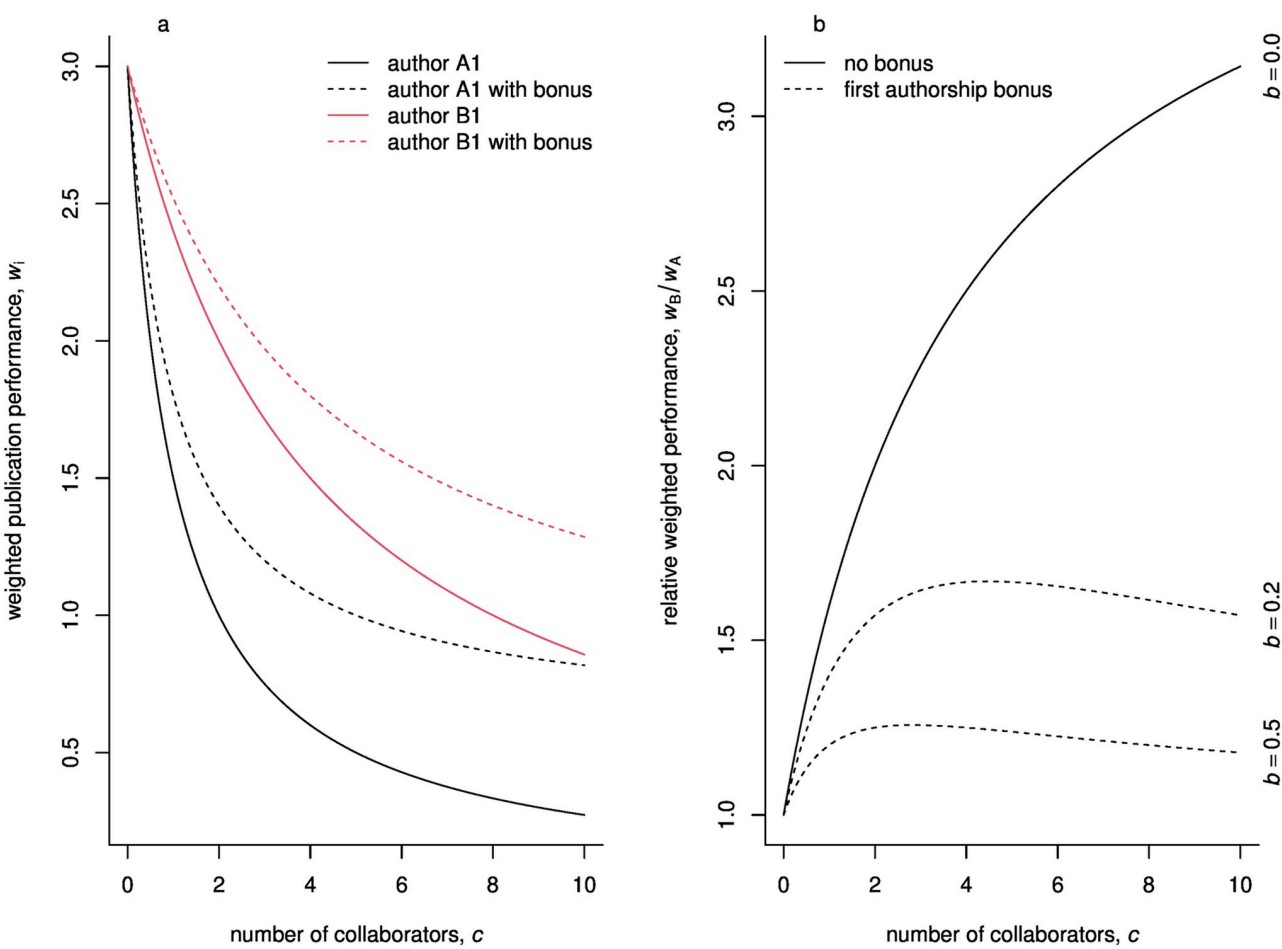

**Fig 2. Publication performance when authors collaborate with people from outside of their groups.** Weighted publication performance of authors $A_1$ and $B_1$ (a). Weighted publication performance of author $B_1$ relative to that of author $A_1$ (b). The weighted publication performance is calculated by taking into account the number of coauthors. During this calculation first authorship can be rewarded by a bonus, $b$. If $b = 0$, then each coauthors receive the same weight for a given publication. On the other hand, if $b > 0$, the weight of the first author is higher than that of the coauthors, i.e. the first author of a paper is rewarded. On subpanel (a) $b = 0.2$, on (b) $b$ is given on the right margin.

For author $A_1$:

$$w_A = \sum_{i=1}^{p} \frac{1}{1+c} = \frac{p}{1+c}.$$

For author $B_1$:

$$w_B = \sum_{i=1}^{Gp} \frac{1}{G+c} = \frac{Gp}{G+c}.$$

The weighted number of papers produced by author $B_1$ relative to author $A_1$, $w_B/w_A$, is:

$$\frac{w_B}{w_A} = \frac{\frac{Gp}{G+c}}{\frac{p}{1+c}} = \frac{Gp}{G+c} \times \frac{1+c}{p} = \frac{G(1+c)}{G+c} = \frac{G+Gc}{G+c}.$$

The proportion of $w_B/w_A$ is greater than one if $G + Gc > G + c$, which is always true if $c > 0$ (as we already assumed $G > 1$, Fig 2). This means that if authors collaborate anyone from outside of their groups then authors in group $B$ will always have higher publication performance than authors in group $A$, despite the fact that the two groups have the same productivity.

To compensate for this productivity bias, author $A_1$ should produce $w_B/w_A$ times more papers, $p_A = p_B(G + Gc)/(G + c)$. This surplus of papers needed for compensating the productivity bias increases with $c$ and it keeps to $G$.

Authors in group $A$ can also compensate for the productivity bias by decreasing the number of their collaborators from outside of the group. This reduction must be by a factor of $G$: $c_A = c_B/G$

A useful modification to the $1/n$ rule is the so called *first-author-emphasis* scheme [18]. In this scheme, the first authors receive a bonus, $b$, to recognise their leading role in producing the papers. Under this scheme the weighted publication performance for author $A_1$, $w'_A$, is:

$$w'_A = \sum_{i=1}^{n_A} \left( b + \frac{1-b}{1+c_A} \right) = \sum_{i=1}^{p} \left( b + \frac{1-b}{1+c} \right) = \frac{p(1+bc)}{1+c}.$$

Here, the first author, who is author $A$ for all his papers, get a bonus $b$ for contributing most to the paper, and the rest of the credit, $1 - b$, is divided equally between all authors (including the first author, [18]). The weighted publication performance for author $B_1$ under the first author scheme, $w'_B$, is:

$$w'_B = \sum_{i=1}^{p_B} \left( b + \frac{1-b}{G_B + c_B} \right) + \sum_{i=1}^{(G_B-1)p_B} \frac{1-b}{G_B + c_B},$$

where the first term gives the credit for first author papers, while the second one is for the coauthored papers. After simplification, we obtain:

$$w'_B = \frac{p(G+bc)}{G+c}.$$

By comparing $w'_B$ to $w'_A$ it is easy to show that author $B_1$ will always have a higher publication performance than author $A_1$, i.e. $w'_B/w'_A > 1$, if $G > 1$ and $b < 1$. Further analysis,

$$\frac{w'_B}{w'_A} = \frac{p(G+bc)}{G+c} \times \frac{1+c}{p(1+bc)} = \frac{G+c[G+b(1+c)]}{G+c[1+b(G+c)]},$$

shows that for $w'_B/w'_A > 1$, the condition $c > 0$ should also be fulfilled. As numerical

computation indicates (Fig 2) the bias is decreased by introducing the first authorship bonus, but it is still significant. The paper [18], for instance, recommend a bonus of $b = 0.2$, but in this case author $B_1$ sill has around 50% more credit for the same work than author $A_1$ has. The difference between authors $A_1$ and $B_1$ decreases as $b$ increases (Fig 2b), but this way coauthorship is worth less and less, undermining the possible benefits of collaborations.

To summarise, this simple model shows that the formation of publication cartels can be an advantageous, but unethical, strategy to increase publication productivity even if one control for the number of coauthors of papers. Note, however, that this model might be overly simplified as all authors have the same primary productivity and we do not investigated how productivity of authors outside of the cartels changes as a consequence of founding cartels. To obtain a more realistic understanding of publication cartels next I develop a simulation of the publication process.

## The simulation

We start simulating the publication process by constructing a publication matrix of papers and authors, $M_P$ (Fig 3). Element $a_{ij}$ of $M_P$ is one if author $j$ is on the bylist of paper $i$ and zero otherwise. Therefore, $M_P$ can be considered as a matrix representation of a bipartite graph, where rows and columns represent the two types of nodes, papers and authors, respectively. To construct $M_P$ we consider a community of $c$ authors. The number of papers written by author $j$ ($j = 1, 2, \ldots, c$) in the community is given by $k_j$. For the community we construct an empty matrix (all $a_{ij} = 0$) of size $p$ and $c$ where $p > \max(k_j)$. Then for each column $j$ we randomly distributed $k_j$ number of ones over the $p$ empty places. Having constructed $M_P$ we create a weighted collaboration (or co-authorship) matrix, $M_C$, by projecting $M_P$ to the nodes of authors. The weights of $M_C$, $J_{ij}$, are Jaccard similarity indices calculated between each pair of authors $i$ and $j$ ($i \neq j$) as

$$J_{ij} = \frac{|P_i \cap P_j|}{|P_i \cup P_j|}.$$

Here, $P_i$ is the set of papers to which author $i$ contributed. In other words, the weight between two authors is the proportion of shared papers to the total number of unique papers to which either of authors $i$ or $j$ contributed to. It varies between zero (i.e. no common publication between author $i$ and $j$) and one (i.e. all publications by the two authors are shared). Note, Jaccard similarity between authors in group A of the above model is zero, while between authors in group B is one. From $M_C$ we construct a collaboration graph, $G_C$.

After creating a random publication matrix I simulated the formation of cartels as follows (Fig 4). First, I choose several authors to form the set $\kappa$ which is the set of authors from the community who form the cartel (i.e. the cartel members). The size of the cartel is given by $|\kappa|$. Then, with probability $p_c$, I changed each element $a_{ij} = 0$ of $M_P$ to $a_{ij} = 1$ where the following conditions met: $j \in \kappa$ and at least one $a_{ik} = 1$ with $k \in \kappa$ but $k \neq j$. I project the resulting publication matrix, $M'_P$ to $M'_C$ and constructed the corresponding collaboration graph, $G'_C$.

The construction of publication networks and formation of cartels were repeated for 1000 times for a given set of parameter values. After constucting the graphs and adding the cartels I calculated the number of papers and the weighted number of papers for each author in the community without and with cartel formed for each repetition. Then these measures were averaged for each author across the 1000 repetitions. To investigate the effects of cartel formation I compare these averaged measures without and with cartel formations. Simulation was

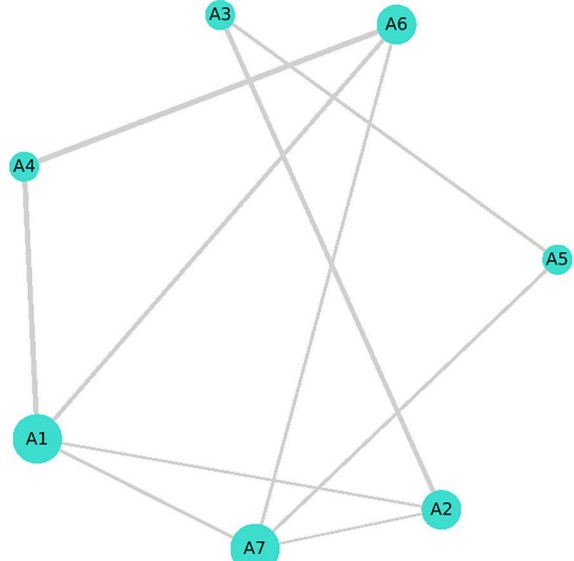

$$J_{A2,A3} = \frac{|A2 \cap A3|}{|A2 \cup A3|} = \frac{2}{5} = 0.4$$

| | A1 | A2 | A3 | A4 | A5 | A6 | A7 |
|---|---|---|---|---|---|---|---|
| A1 | . | 0.2 | . | 0.5 | . | 0.33 | 0.25 |
| A2 | . | . | 0.4 | . | . | . | 0.17 |
| A3 | . | . | . | 0.25 | . | . | . |
| A4 | . | . | . | . | . | 0.5 | . |
| A5 | . | . | . | . | . | . | 0.25 |
| A6 | . | . | . | . | . | . | 0.25 |
| A7 | . | . | . | . | . | . | . |

**Fig 3. The construction of publication network.** The top left panel shows the publication matrix, $M_P$. Each row and column of this matrix represents a paper and an author, respectively. Values of 1 indicate that an author is on the author list of a given paper, while dots symbolise zeros. From the publication matrix one can derive the collaboration matrix, $M_C$ (bottom right panel) by calculating the Jaccard simmilarity (top right) for each possible pairs of authors. The bottom left panel shows the resulting weighted, undirected collaboration graph, $G_C$. Node size is proportional to the number of coauthors (degree), while edge width shows the strength of the connection between two authors (i.e. it is proportional to their Jaccard similarity). The red rectangles in the matrices exemplify the calculation of Jaccar simmilarity.

implemented in the Julia programming language [19], and available on GitHub (https://github.com/zbartab/PaperPump).

By setting all $k_j = k$ and $p \gg k$ we can simulate the case of equal productivity and no collaboration from outside of the group. Here the simulation produces the same results as the model: productivity of cartel members increased but this can be accounted for by using the weighted number of publications (result not shown).

To induce collaboration between authors I next set $p < \sum_{j=1}^{c} k_j = ck$ (the authors still have the same productivity prior to cartel formation). Under these conditions, if we consider the

|    | A1 | A2 | A3 | A4 | A5 | A6 | A7 |
|----|----|----|----|----|----|----|----|
| p1 | ·  | 1  | ·  | ·  | ·  | ·  | ·  |
| p2 | ·  | ·  | ·  | ·  | ·  | 1  | 1  |
| p3 | ·  | ·  | ·  | ·  | 1  | ·  | 1  |
| p4 | ·  | 1  | 1  | ·  | ·  | ·  | ·  |
| p5 | 1  | ·  | ·  | 1  | ·  | 1  | ·  |
| p6 | ·  | 1  | 1  | ·  | ·  | ·  | ·  |
| p7 | ·  | ·  | 1  | ·  | 1  | ·  | ·  |
| p8 | 1  | 1  | ·  | ·  | ·  | ·  | 1  |

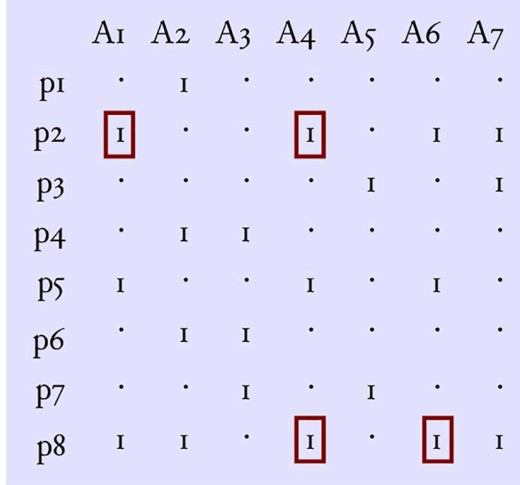

|    | A1 | A2 | A3 | A4 | A5 | A6 | A7 |
|----|----|----|----|----|----|----|----|
| p1 | ·  | 1  | ·  | ·  | ·  | ·  | ·  |
| p2 | [1] | · | ·  | [1] | ·  | 1  | 1  |
| p3 | ·  | ·  | ·  | ·  | 1  | ·  | 1  |
| p4 | ·  | 1  | 1  | ·  | ·  | ·  | ·  |
| p5 | 1  | ·  | ·  | 1  | ·  | 1  | ·  |
| p6 | ·  | 1  | 1  | ·  | ·  | ·  | ·  |
| p7 | ·  | ·  | 1  | ·  | 1  | ·  | ·  |
| p8 | 1  | 1  | ·  | [1] | ·  | [1] | 1  |

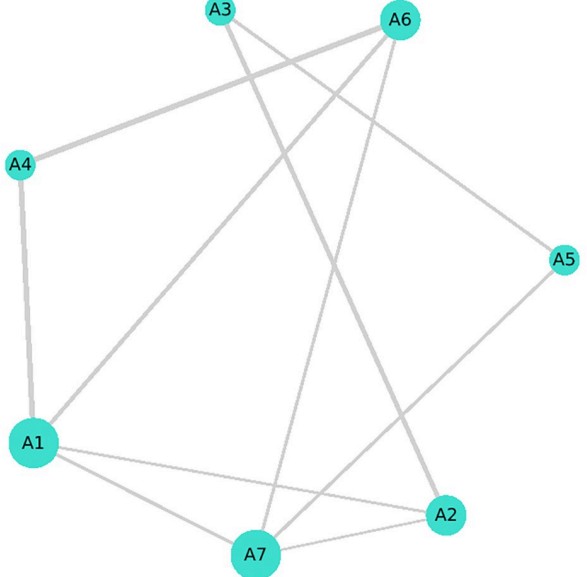

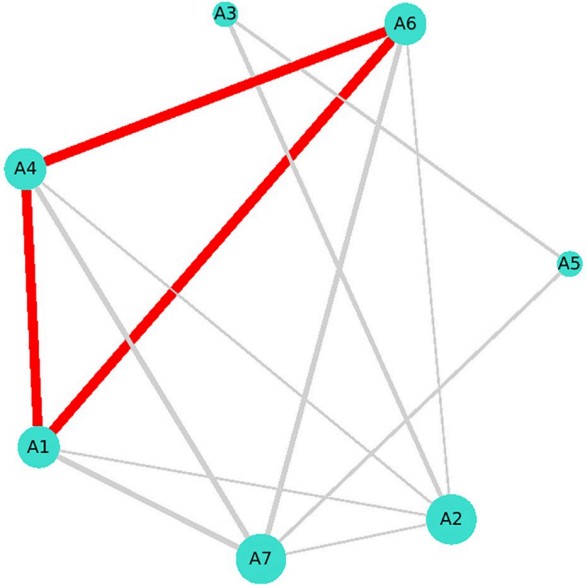

**Fig 4. The formation of cartels.** The panels on the left illustrate a publication network without cartel. The panels on the right show how a cartel between authors A1, A4 and A6 can be formed: Author A6 invites authors A1 and A4 to be coauthors on paper p2, while author A1 do the same with authors A4 and A6 on paper p8. The small red rectangles mark the authorships gained this way. The bottom right panel shows the resulting collaboration graph. Node size is proportional to the number of coauthors (degree), while edge width shows the strength of the connection between two authors (i.e. it is proportional to their Jaccard similarity). The red edges connect cartel members. Note (i) the strong connections between members and (ii) adding cartels also changes the connections of non-members.

number of papers, the productivity of cartel members increases significantly by forming cartel while productivity of non-members does not change (Fig 5). In accordance with the model, the productivity of cartel members increases even if we consider the weighted number of papers. Interestingly, the productivity of non-members decreases when cartel is formed (Fig 5).

I further generalise the simulation results by setting the prior productivity of authors to different values (Fig 6). Using the number of papers as metric leads to the same conclusions: members' productivity increases after cartel formation, non-members' productivity does not

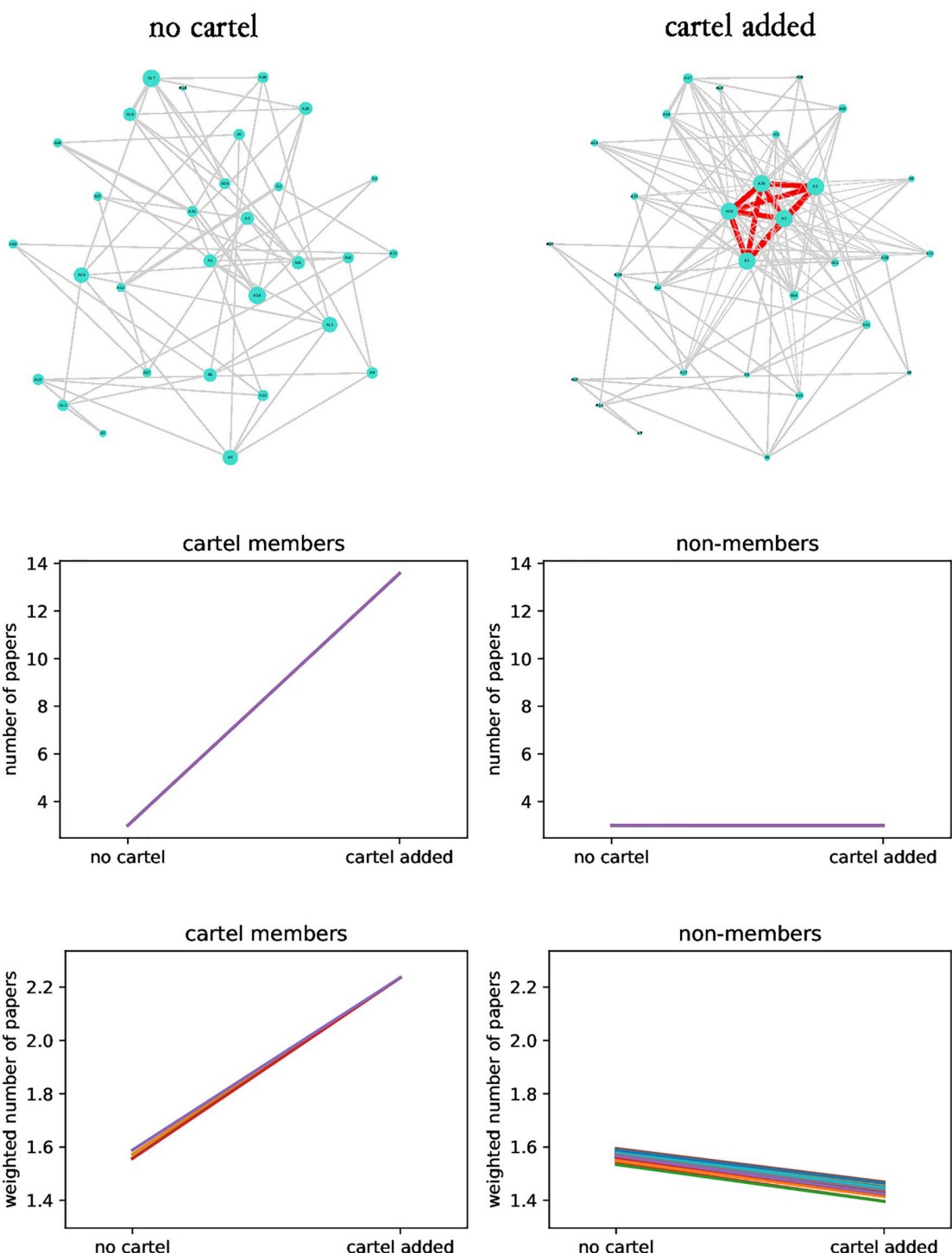

**Fig 5. The effect of cartel formation on the productivity of cartel members and non-members: Equal prior productivity of authors.** The top panels illustrate the collaboration graph without and with cartel formation. The middle panels show how the number of papers produced by members and non-members changes because of founding cartel. The bottom panels illustrate the same but using the weighted number of papers as a measure of productivity. Collaboration graphs were formed with $c = 30$, $k = 3$, $p = 60$, $p_c = 1$ and $\kappa = \{1, 2, 3, 29, 30\}$ (cartel composition is arbitrary as all authors are the same in terms of prior productivity). Averages of 1000 repetitions are plotted and different colours represent different authors.

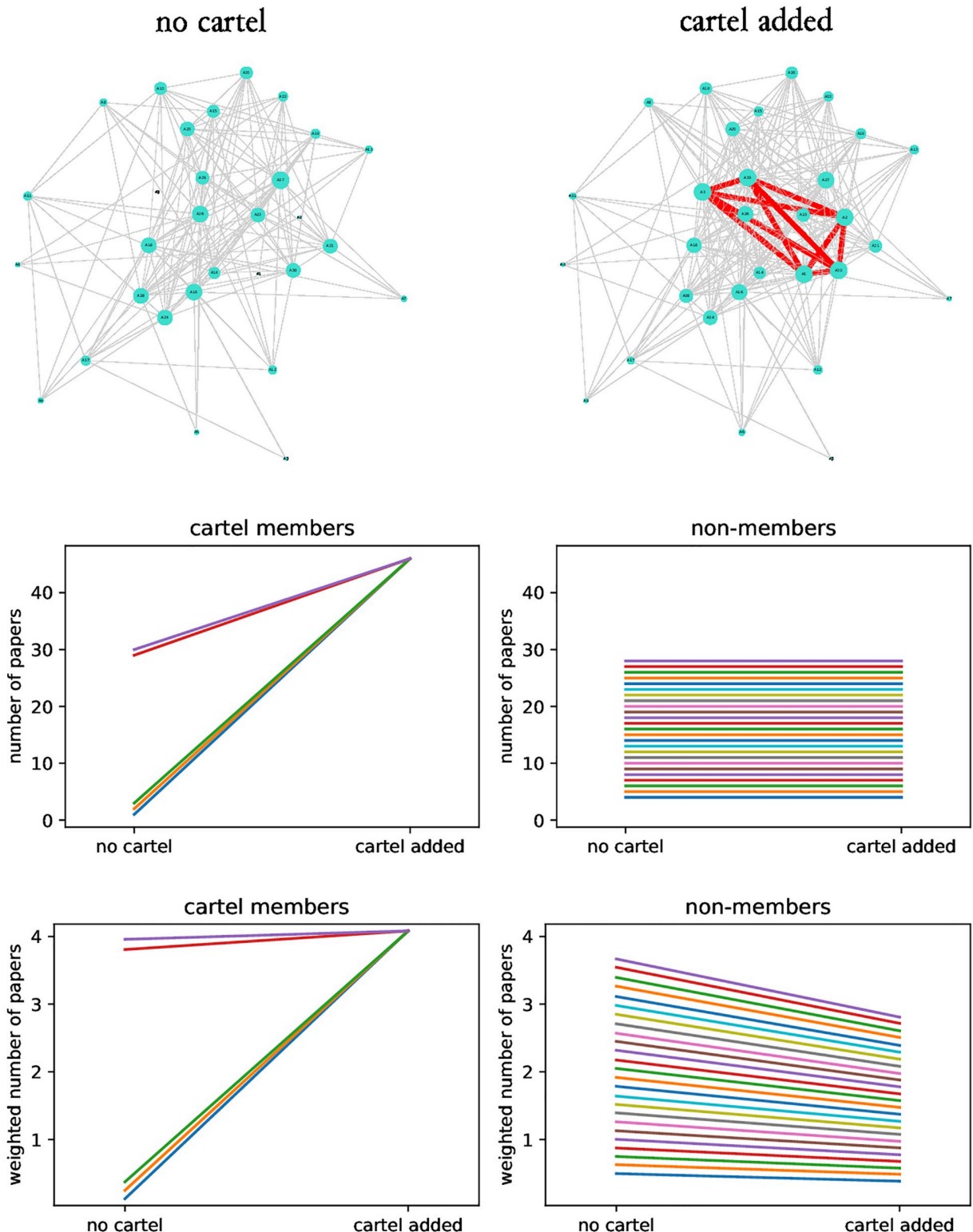

**Fig 6. The effect of cartel formation on the productivity of cartel members and non-members: Prior productivity of authors differs.** The top panels illustrate the collaboration graph without and with cartel formation. The middle panels show how the number of papers produced by members and non-members changes because of founding cartel. The bottom panels illustrate the same but using the weighted number of papers as a measure of productivity. Collaboration graphs were formed with $c = 30$, $k_j = j$, $p = 60$, $p_c = 1$ and $\kappa = \{1, 2, 3, 29, 30\}$ (cartel composition illustrate the case when authors of very low and very high productivity form a cartel). Averages of 1000 repetitions are plotted and different colours represent different authors.

change. Using the weighted number of papers, similarly to the previous case, productivity of non-members decreases as a consequence of cartel formation. Cartel members' productivity increases with cartel formation but this increase is uneven: members with low prior productivity have a significant increase, while the increase for members with high prior productivity is marginal (Fig 6). This suggests that different individuals migh benefit differently from cartel formation.

To investigate individual differences further I randomly created 1000 cartels and, as above, repeat the simulation of publication process with each of these cartels for 100 times. From these simulations I calculated the difference in weighted numbers of papers between with and without cartels for each cartel member, averaged over the repetitions of each cartel. This value represents the effect of cartel formation, i.e. how an individual's weighted number of publication would change if it participate in the given cartel compared to the case of no cartel formation. I characterised cartels by the meand and standard deviation (SD) of their members' prior productivity, $k_j$. Low mean indicates cartel formed by low productivity individuals, while high means signal just the opposite. Cartels with low SD represent uniform group of individuals, i.e. everybody has similar prior productivity, while high SD means diverse cartels where the members' prior productivity are largely different.

As Fig 7 (left panel) shows the change in weighted number of papers increases with the mean prior productivity of the cartel, i.e. cartel formation is most benefitial for a given individual if its cartel partners are highly productive. Members' gain, however, is far from equal, individuals with low prior productivity benefit most from cartel formation. Individuals with high prior productivity can even loss with carter formation if their partners' mean prior

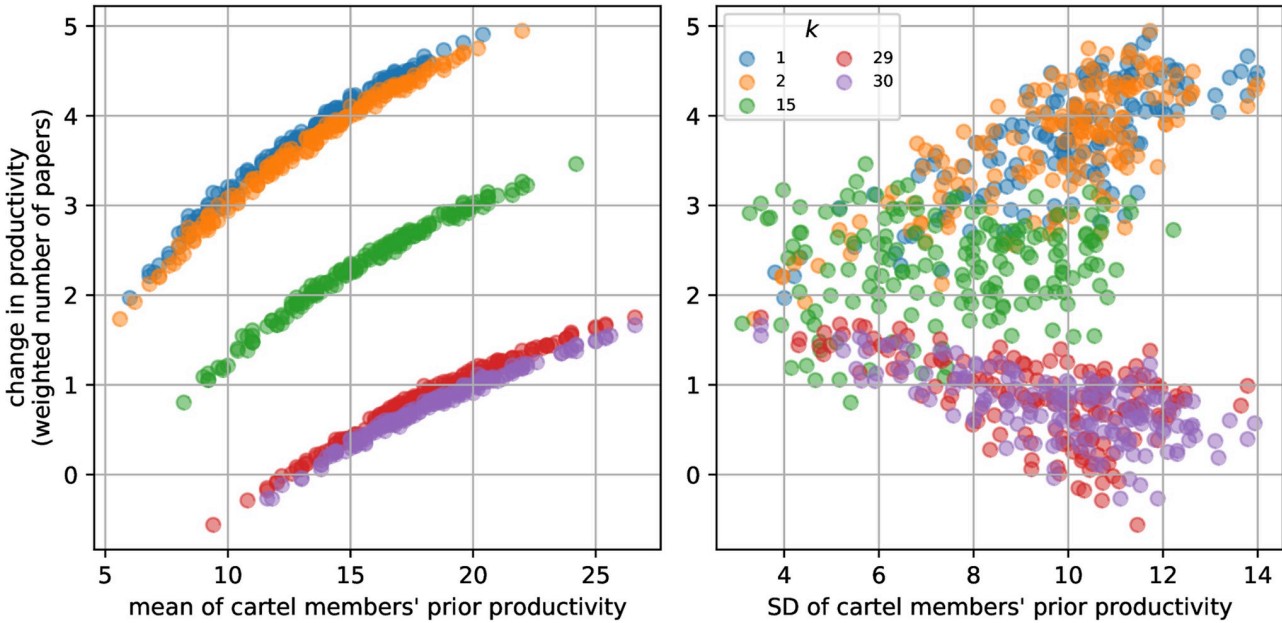

**Fig 7. Effect of cartel formation on productivity for differently productive cartel members.** The left panel shows the difference in weighted number of papers between simulation with and without cartels for individuals with different prior productivity (see legend on the right panel) as a function of mean prior productiviy of cartel members. Note the negative values (i.e. cartel formation decreases productivity) for highly productive members ($k$ = 29, 30) at low average cartel productivity. The right panel show the same change as a function of cartel diversity (measured as the standard deviation (SD) of members' prior productivity). Note, the effect of cartel formation increases with cartel diversity for individuals of low prior production, while decreases for highly productive members (it can be even negative at high cartel diversity). Collaboration graphs were formed with $c$ = 30, $k_j$ = $j$, $p$ = 60, $p_c$ = 1 and cartels randomly formed. Each point is the average of 100 simulation for the same cartel. For clarity, only data for authors with extreme low and high and intermediate prior productivity are shown.

productivity is low (note the negative values of change for individuals with $k = 29, 30$). The diversity of cartels influences the benefit of cartel formation differently for the different individuals (Fig 7, right panel). Cartel members of low productivity gain more and more as the diversity of their cartel increases. If a low productivity individual finds itself in a diverse cartel that neccessarily means that it is teamed up with highly productive individuals who produce many papers authorship on which for the low productivity individuals can be gifted. For individuals with intermediate productivity cartel diversity has no effect. On the other hand, highly productive individuals can even loss in diverse cartels. When a highly productive individual is in a diverse cartel then most of its partners are of low productivity whose papers cannot contribute significantly to the gain of the high productivity author.

## Conclusion

Under the current climate of wide spread use of scientometry indices to assess academics publication cartels can provide huge, although unethical benefits. As my results indicate, members of cartels by reciprocally inviting each other as honorary authors can easily boost their own publication productivity, i.e. the number of papers they appear on as (co)author. As many scientometrics currently in use are strongly associated with the number of publications a scholar has produced [3, 8, 9] becoming cartel member can have a very general positive effect on one's academic career.

One may consider that fighting off cartels is not necessary because of "no harm no foul": research integrity may not be inevitably damaged by cartel foundation, cartels can produce high quality research. Nevertheless, cartels do distort the research competition landscape. This might result in that highly competent, talented researchers, who are not members of any cartels, are forced into inferior roles which, in turn, compromises the society's ability to produce more novel and innovative results. Therefore, cartel formation should be restricted.

Fighting against cartels is, however, not trivial. First, identifying cartels, not to mention to prove for a group of researchers that they are cartelling is inherently difficult. Investigating properties of coauthor networks might help as indicated here by the strong connections among cartel members in the simulated collaboration networks. Nevertheless, a possible way to restrict cartels without their identification is to use such scientometrics which penalise cartel formation. An obvious choice can be to weight the number of publications an author has by the inverse of the number of authors on the bylists of these papers, the so-called $1/n$ rule [12, 18]. As my calculation shows this rule can only be effective if coauthorship only occurs between cartel members. As soon as collaboration is wide spread among both cartelling and non-cartelling authors my results indicate that the $1/n$ rule breaks down and cartel members still gain undeserved benefits. On the other hand, my computations also show that the $1/n$ rule can still be useful against publication cartels, because it generates conflicts of interest among the parties. Collaborators of cartel members suffer a loss if the $1/n$ rule is applied which might force them either to change the unethical behaviour of cartel members or abandon to collaborate with them.

The computations also show, given that the $1/n$ rule is used to rate scholars, that authors of different productivity should persuade different strategy when forming/joining cartels. Low productivity authors are doing best by being alone in a cartel of high productivity authors. On the other hand, for profilic authors the best strategy is to form cartels among themselves, because cartel establishment with lowly ranked authors can have a detrimental effect on their productivity. As highly productive scientists can be assumed to have more power than their low productivity fellows, i.e. they are able to exlude weakly performing authors from among themselves, it is expected that cartels are formed by authors of similar prior productivity under

the $1/n$ rule. The results also suggest that scholars of low productivity gain the most by cartel formation, therefore founding cartels might be the most common among them. Although, profilic authors might also have an interest to form cartels to avoid being overtaken by cartel forming lower productivity authors. As these arguments suggest the introduction of $1/n$ rule for researcher assessment can generate a dynamic publication landscape where several conflicts of interest can arise. This is very different from the current assessment scheme where everybody's interest is the same. Currently, it is not entirely clear if this dynamism can help fight agaïnts cartel formation.

To summarise, I strongly argue for using the $1/n$ rule as the basis of scientometry. Unfortunately, its general use is opposed by many parties for many reasons. It still remains to see whether these reasons are valid or not, but my calculations indicate that the application of $1/n$ rule can generate such processes which may ultimately lead to the self-purification of the academic publication industry. Of course, abandoning the current, metric-only research assessment system can also help.

## Acknowledgments

I thank Miklós Bán, Gábor Lövei, Tibor Magura, Jácint Tökölyi and an anonymous referee to review a previous version of the manuscript.

## Author Contributions

**Conceptualization:** Zoltán Barta.

**Formal analysis:** Zoltán Barta.

**Funding acquisition:** Zoltán Barta.

**Investigation:** Zoltán Barta.

**Methodology:** Zoltán Barta.

**Project administration:** Zoltán Barta.

**Resources:** Zoltán Barta.

**Software:** Zoltán Barta.

**Supervision:** Zoltán Barta.

**Validation:** Zoltán Barta.

**Visualization:** Zoltán Barta.

**Writing – original draft:** Zoltán Barta.

**Writing – review & editing:** Zoltán Barta.

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
