## [Decision Letter · Decision Letter 0]

26 Jul 2022

PONE-D-22-16973Publication games: in the web of reciprocityPLOS ONE

Dear Dr. Barta,

Thank you for submitting your manuscript to PLOS ONE. After careful consideration, we feel that it has merit but does not fully meet PLOS ONE’s publication criteria as it currently stands. Therefore, we invite you to submit a revised version of the manuscript that addresses the points raised during the review process.

Reviewer #1 suggests that the paper should be accepted. I directly read the paper and I think that it should be slightly revised before publication. 

1) The paper omitted completely the technicalities of simulations. I think that you should add information about the technique and code you used. This probably led to some difficulties in reading Figure 5 and Figure 6.

2) In all the figures, the size of node changes. I assume it is proportional to the weighted number of papers authored by a node author. Moreover, the use of different colours in the lower panels of figures 5- and 6 is not explicitly addressed.  Finally, the choice of representing a specific cartel in the graph is not commented at all.

3) There are some minor inconsistencies in the notation (use of ’ instead of  ').

4) As for discussion and policy indication, in I understand correctly your analysis, the suggestion that the adoption of the rule of 1/n may lead to self-purification is not consistent with your results.. The conflict of interest induced by the rule is valid only for authors of different productivity.  As you explicitly stated, if a group of similar productivity authors forms a cartel, it boosts the productivity of its members. Hence, there is a clear strategy that similar-productivity authors may adopt for gaining positions in the publication game. Moreover, in a dynamic game, where people of similar productivity gain position, also the higher-productivity authors may have an interest to form cartels in view of avoiding being reached by cartels formed by lower-productivity authors.     Please submit your revised manuscript by Sep 09 2022 11:59PM. If you will need more time than this to complete your revisions, please reply to this message or contact the journal office at plosone@plos.org. Please include the following items when submitting your revised manuscript:A rebuttal letter that responds to each point raised by the academic editor and reviewer(s). You should upload this letter as a separate file labeled 'Response to Reviewers'.A marked-up copy of your manuscript that highlights changes made to the original version. You should upload this as a separate file labeled 'Revised Manuscript with Track Changes'.An unmarked version of your revised paper without tracked changes. You should upload this as a separate file labeled 'Manuscript'.

We look forward to receiving your revised manuscript.

Kind regards,

Alberto Baccini, Ph.D.

Academic Editor

PLOS ONE

Journal Requirements:

Reviewers' comments:

Reviewer's Responses to Questions

**Comments to the Author**

1. Is the manuscript technically sound, and do the data support the conclusions?

Reviewer #1: Yes

2. Has the statistical analysis been performed appropriately and rigorously? 

Reviewer #1: Yes

3. Have the authors made all data underlying the findings in their manuscript fully available?

Reviewer #1: Yes

4. Is the manuscript presented in an intelligible fashion and written in standard English?

Reviewer #1: Yes

5. Review Comments to the Author

Reviewer #1: The paper deals with an important distortion in the production of science, which is the formation of cartel publications. Besides individual incentives associated with higher productivity leading to higher salaries and promotion, there are also institutional incentives to push faculty to increase their research output, such as public founding agencies and/or college associations membership.

See, for example, Besancenot et al. (2009) Why Business Schools do so much research: A signaling Explanation (2009) Research Policy 38, 1093-1101

Faria, J. and F. Mixon (2022) Opportunism vs. excellence in academia: Quality accreditation of collegiate business schools (2022) American Business Review, open source.

6. PLOS authors have the option to publish the peer review history of their article (what does this mean?). If published, this will include your full peer review and any attached files.

Reviewer #1: No

---

## [Author Response · Author response to Decision Letter 0]

20 Sep 2022

Alberto Baccini, Ph.D. 

Academic Editor 

PLOS ONE 

__RE: PONE-D-22-16973, "Publication games: in the web of reciprocity"__

Dear Dr. Baccini,

Many thanks for effort to deal with my manuscript and your overall positive opinion. Below I respond to all of your and the reviewer's comments (typed in italic) in details. Line numbers corresponds to the marked-up copy of the MS.

_1) The paper omitted completely the technicalities of simulations. I think that you should add information about the technique and code you used. This probably led to some difficulties in reading Figure 5 and Figure 6._

Thank you for pointing out this deficiency in the MS. Now I added a paragraph (l. 171-179) about the simulations. During this I recognised that simulations were performed only once so I repeated them 1000 times. As a result my conclusion on Fig 6 changed a bit (l. 194-202), so I performed more simulations to clarify this which resulted in a new figure (Fig 7) and some more text (l. 203-231). These results show that the application of 1/n rule can have different effect of authors of different prior productivity. I now discuss these findings in the Conclusions (l. 263-286).

_2) In all the figures, the size of node changes. I assume it is proportional to the weighted number of papers authored by a node author. Moreover, the use of different colours in the lower panels of figures 5- and 6 is not explicitly addressed. Finally, the choice of representing a specific cartel in the graph is not commented at all._

You are right again here. I added to the figure legends that node size is proportional to the number of coauthors (degree). Use of different colours are also clarified in the legends of Fig 5 and 6 as well as the use of specific cartels are justified here.

_3) There are some minor inconsistencies in the notation (use of ’ instead of ')._

These are corrected now.

_4) As for discussion and policy indication, in I understand correctly your analysis, the suggestion that the adoption of the rule of 1/n may lead to self-purification is not consistent with your results.. The conflict of interest induced by the rule is valid only for authors of different productivity. As you explicitly stated, if a group of similar productivity authors forms a cartel, it boosts the productivity of its members. Hence, there is a clear strategy that similar-productivity authors may adopt for gaining positions in the publication game. Moreover, in a dynamic game, where people of similar productivity gain position, also the higher-productivity authors may have an interest to form cartels in view of avoiding being reached by cartels formed by lower-productivity authors._

You are absolutely right here, I modified the Conclusions accordingly taking into account the new results presented in Fig 7 as well.

_Reviewer #1: The paper deals with an important distortion in the production of science, which is the formation of cartel publications. Besides individual incentives associated with higher productivity leading to higher salaries and promotion, there are also institutional incentives to push faculty to increase their research output, such as public founding agencies and/or college associations membership. See, for example, Besancenot et al. (2009) Why Business Schools do so much research: A signaling Explanation (2009) Research Policy 38, 1093-1101 Faria, J. and F. Mixon (2022) Opportunism vs. excellence in academia: Quality accreditation of collegiate business schools (2022) American Business Review, open source._

Thank you for pointing out these wider implications. Faria & Mixon is referenced now to call attention for this phenomenon too.

I hope that after these modifications the MS is now suitable for publication in PLOS ONE.

Sincerely yours, 

Zoltan Barta

---

## [Decision Letter · Decision Letter 1]

11 Oct 2022

Publication games: in the web of reciprocity

PONE-D-22-16973R1

Dear Dr. Barta,

We’re pleased to inform you that your manuscript has been judged scientifically suitable for publication and will be formally accepted for publication once it meets all outstanding technical requirements.

Kind regards,

Alberto Baccini, Ph.D.

Academic Editor

PLOS ONE

Additional Editor Comments (optional):

Reviewers' comments:

Reviewer's Responses to Questions

**Comments to the Author**

1. If the authors have adequately addressed your comments raised in a previous round of review and you feel that this manuscript is now acceptable for publication, you may indicate that here to bypass the “Comments to the Author” section, enter your conflict of interest statement in the “Confidential to Editor” section, and submit your "Accept" recommendation.

Reviewer #1: All comments have been addressed

2. Is the manuscript technically sound, and do the data support the conclusions?

Reviewer #1: Yes

3. Has the statistical analysis been performed appropriately and rigorously? 

Reviewer #1: N/A

4. Have the authors made all data underlying the findings in their manuscript fully available?

Reviewer #1: Yes

5. Is the manuscript presented in an intelligible fashion and written in standard English?

Reviewer #1: Yes

6. Review Comments to the Author

Reviewer #1: n/a

7. PLOS authors have the option to publish the peer review history of their article (what does this mean?). If published, this will include your full peer review and any attached files.

Reviewer #1: No

---

## [Editor Report · Acceptance letter]

17 Oct 2022

PONE-D-22-16973R1 

Publication games: in the web of reciprocity 

Dear Dr. Barta:

I'm pleased to inform you that your manuscript has been deemed suitable for publication in PLOS ONE. Congratulations! Your manuscript is now with our production department. 

Kind regards, 

on behalf of

Prof. Alberto Baccini 

Academic Editor

PLOS ONE